# Muscular Performance Is Not Significantly Altered Throughout Phases of the Menstrual Cycle or a Hormonal Contraceptive Cycle in Collegiate Softball Players

**DOI:** 10.3390/muscles4030037

**Published:** 2025-09-02

**Authors:** Shelby L. Houchlei, Sarah N. Wood, Sarah E. Peters, Shane K. Miller, Taylor K. Dinyer-McNeely, Ryan A. Gordon

**Affiliations:** 1Exercise Physiology Laboratory, McDonald Arena, Missouri State University, Springfield, MO 65897, USA; sh986s@missouristate.edu (S.L.H.); tdinyer-mcneely@missouristate.edu (T.K.D.-M.); 2School of Health Sciences, Missouri State University, Springfield, MO 65897, USA

**Keywords:** female, athletes, muscular performance, performance testing, strength, sprint

## Abstract

Potential variability in neuromuscular function or physiology throughout the menstrual cycle (MC) or a cycle of using hormonal contraceptives may affect muscular performance variables that are relevant to exercise, training, or sport. Collegiate softball players (n = 11) that reported using and not using hormonal contraceptives completed three testing sessions during their respective early follicular, ovulatory, and mid luteal phases of the MC or early, mid, or late phases of their hormonal contraceptive cycle (HCC). Each testing session included a series of performance tests: countermovement jump on a force plate, 15-yard sprints, velocity assessment of the back squat performed at 70% of one-repetition maximum (1-RM), one-repetition maximum bench press, and 70% 1-RM repetitions to failure testing on the bench press. No significant differences were found for any of the performance tests between the three phases, though performance on most tasks peaked during the mid luteal/late phases of the MC/HCC. It is important to note that this study was underpowered and this could have masked any observed differences. Collectively, muscular performance was not significantly different throughout phases of the MC or HCC in these athletes, indicating that potential hormonal variability throughout the MC or HCC did not seem to have an effect on performance outcomes in this study.

## 1. Introduction

The menstrual cycle (MC) is characterized by cyclical fluctuations in female sex hormones [1,2]. In eumenorrheic females, the MC lasts approximately 21–35 days [1]. It is categorized into two main phases: the follicular and luteal phases, which can be further divided into six subphases: early follicular (EF), late follicular (LF), ovulatory (OV), early luteal (EL), mid luteal (ML), and late luteal (LL), with each phase of the MC lasting approximately 3–7 days [1]. These sub-phases are characterized by fluctuations in sex hormones, including estrogen and progesterone, along with follicle stimulating hormone and luteinizing hormone. At the onset of a new cycle (EF), estrogen concentrations are low, and this low concentration is kept constant throughout EF, whereby it gradually increases, peaks during LF and remains elevated throughout OV. This is then followed by an overall reduction in estrogen throughout the luteal phase, whereas low progesterone concentrations are kept constant throughout the follicular phase. This is followed by an increase during OV, before peaking during EL/ML. Towards the end of the luteal phase, progesterone rapidly declines, which leads into menstruation and the onset of a new MC [1,2,3]. In women using hormonal contraceptives, the release of follicle stimulating hormone and luteinizing hormone is halted due to disruptions in the cyclical changes in estrogen and progesterone [4,5]. With this, phases and sub-phases that are typically observed as part of the MC differ in women using hormonal contraceptives and can be considered instead as a cyclical time period of using hormonal contraceptives (i.e., hormonal contraceptive cycle or HCC as opposed to an MC) [5].

Estrogen and progesterone are proposed to have a variety of profound effects on neuromuscular physiology, including neural activation of muscle, the adaptive potential of muscle to training, muscle metabolism and protein synthesis, and its reparative capabilities [6,7,8]. Considering this, fluctuations in estrogen and progesterone throughout an MC or an HCC could result in variability in muscular performance variables such as force or power production at different phases of an MC or HCC [1,9]. Previous studies have shown that estrogen has an excitatory effect on neural activity, while progesterone has an inhibitory effect on neural activity, reinforcing the belief that neuromuscular activity and physiology may vary given the fluctuations in both hormones that can occur throughout an MC or HCC [1,10,11]. Thus, neuromuscular activity and physiology may be augmented during phases of the MC or HCC that are characterized by elevated estrogen and low progesterone concentrations, such as LF and OV phases of the MC, which may translate into enhanced performance characteristics such as force or power [1].

In their systematic review and meta-analysis, McNulty et al. reported a slight reduction in exercise performance in the early follicular phase when compared to other phases of the MC, but it was noted that the effect size was often small and there was a high degree of variability between many of the studies [12]. Similarly, Niering et al. reported different expressions of maximal strength (isometric, isokinetic, dynamic) were higher during LF and OV compared to EF, though the effect sizes were small to medium [13]. These two systematic reviews provide a good-sized body of evidence showing exercise performance and muscular performance can vary at different phases of the MC, though the magnitude of variation appears to be relatively small. Despite this, our understanding of how muscle-centric variables such as force and power deviate throughout the MC or an HCC is still unclear, particularly in female athletes participating in high-stakes training and competition. To address this, the present study measured muscular performance through a series of assessments at three phases of the MC (EF, OV, and ML) or at three similar timepoints of an HCC (equivalent to MC; early, mid, and late) in collegiate female athletes that were both users and non-users (eumenorrheic) of hormonal contraceptives. We hypothesized that participants’ performance on the assessments would peak during the OV phase and mid-timepoint phase of the MC and HCC, respectively.

## 2. Results

Eleven (n = 11) participants completed this study, and their descriptive characteristics are presented in Table 1. Descriptives are separated in Table 1 for users of hormonal contraceptives (Con), non-users (Non), as well as all participants combined. Participants in Non were assessed during each athlete’s EF, OV, and ML phases of the MC, whereas Con were assessed during a time period (e.g., day five) of their respective HCC that was equivalent to Non using self-report and day-counting methods. Given that Non and Con were assessed over similar timelines according to their MC or HCC, and for the sake of simplicity, results for all participants are classified into three phases: EF (early follicular or early phase of HCC), OV (ovulatory or mid phase of HCC), and ML (mid luteal or late phase of HCC.

### 2.1. Countermovement Jump Performance During EF, OV, and ML

There were no differences in relative max force (N/kg; EF = 17.1, OV = 16.7, ML = 17.1, *p* = 0.622), relative max power (W/kg; EF = 28.9, OV = 28.0, ML = 27.3, *p* = 0.711), max velocity (m/s; EF = 2.64, OV = 2.57, ML = 2.61, *p* = 0.409), concentric time (s; EF = 0.34, OV = 0.34, ML = 0.34, *p* = 0.851), eccentric time (s; EF = 0.41, OV = 0.38, ML = 0.40, *p* = 0.815), jump height (m; EF = 0.36, OV = 0.34, ML = 0.35, *p* = 0.403), time to max acceleration (s; EF = 0.99, OV = 0.91, ML = 0.91, *p* = 0.476), time to takeoff (s; EF = 1.10, OV = 1.03, ML = 1.03, *p* = 0.480), and braking rate of force development (RFD; N/s; EF = 2294.8, OV = 2259.1, ML = 2478.2, *p* = 0.912) during countermovement jump attempts between the EF, OV, and ML phases. Data and effect sizes between phases for each variable from the countermovement jump are shown in Table 2.

### 2.2. Fifteen-Yard Sprint Performance During EF, OV, and ML

There were no differences in peak 15-yd sprint performance between the EF, OV, and ML phases (s; EF = 2.16 ± 0.09, OV = 2.19 ± 0.1, ML = 2.2 ± 0.1, *p* = 0.839, Figure 1A,B). There were small to minimal effect sizes reported when comparing performance between phases (EF vs. OV; ES = −0.35, EF vs. ML; ES = −0.1, OV vs. ML; ES = −0.22).

### 2.3. Back Squat (70% 1RM) Velocity, Bench Press 1RM, and Bench Press (70% 1RM) Repetitions to Failure Performance During EF, OV, and ML

There were no differences in peak velocity for the 70% back squat between the EF, OV, and ML phases (m/s; EF = 0.65 ± 0.08, OV = 0.64 ± 0.07, ML = 0.66 ± 0.07, *p* = 0.871, Figure 2A,B). There were small effect sizes reported when comparing performance between phases (EF vs. OV; ES = 0.13, EF vs. ML; ES = −0.13; OV vs. ML; ES = −0.27). There were no differences in 1RM for the bench press between the EF, OV, and ML phases (kg; EF = 50.4 ± 5.9, OV = 50.7 ± 5.7, ML = 52.3 ± 5.6, *p* = 0.717, Figure 3A,B). There were minimal to small effect sizes reported when comparing 1RM performance between phases (EF vs. OV; ES = −0.05, EF vs. ML; ES = −0.33, OV vs. ML; ES = −0.28). Additionally, there were no differences in maximal repetitions performed (RTF) for the 70% bench press between the EF, OV, and ML phases (repetitions; EF = 11.5 ± 3.2, OV = 12.1 ± 4.0, ML = 12.8 ± 3.3, *p* = 0.645, Figure 4A,B). There were small effect sizes reported when comparing performance between phases (EF vs. OV; ES = −0.16, EF vs. ML; ES = −0.39, OV vs. ML; ES = −0.19).

## 3. Discussion

The purpose of this study was to determine if muscular performance variables that are central to exercise training and sport varied throughout phases of the MC or HCC (EF, OV, and ML) in collegiate female softball athletes, including both users (Con) and non-users (Non) of hormonal contraceptives. Based on the proposed effects of estrogen and progesterone on neuromuscular function [1,10,11], we hypothesized that performance would peak in the OV phase compared to the EF and ML phases. There were no significant differences in any of the performance variables between the three phases, though performance tended to be highest for most variables during ML. Specifically, relative max force (N/kg; equivalent to EF), time to max acceleration (s), time to take-off (s), and braking RFD (N/s) determined from the countermovement jump test, along with performance on the 70% 1RM back squat velocity, 1RM bench press, and bench press RTF tests all peaked during ML. Moreover, relative max force (N/kg; equivalent to ML), relative max power (W/kg), max velocity (s), and jump height (m) during the countermovement jump all peaked during EF. Thus, counter to our hypothesis, performance did not significantly vary throughout the MC, and raw values for these performance variables tended to favor ML as well as EF.

As reported previously, indices of muscular performance appear to be marginally affected throughout the MC and most cycle-related effects on performance are likely to be highly individualized [12]. For instance, McNulty et al. reported minor reductions in exercise performance (median effect size of all studies was −0.06), including measurements of strength and endurance performance, during the EF phase in comparison to the OV and ML phases of the MC in eumenorrheic women not using hormonal contraceptives [12]. A recent meta-analysis reported different expressions of strength (isometric, isokinetic, dynamic) varied across phases of the MC in eumenorrheic women. Across 22 studies, Niering et al. found isometric maximal strength and dynamic strength (e.g., 1RM) peaked during the late follicular phase, whereas isokinetic maximal strength peaked during ovulation, with small to medium effects sizes observed [13]. Julian et al. assessed performance on a series of performance tests during the early follicular and mid luteal phase of the MC in eumenorrheic, competitive female soccer players [14]. Similar to our findings, performance on the Yo-Yo Intermittent Endurance Test varied between the two phases, though this effect was not significant (*p* = 0.07), whereas no differences were observed for jump height during a countermovement jump and 30 m sprint time. Tsampoukos et al. found peak and mean power output during 30-sec sprints to be unaffected when assessed during the follicular, ovulatory, and luteal phases of the MC in eumenorrheic, recreational female athletes [15]. Furthermore, García-Pinillos et al. reported no significant differences in vertical jump, sprint, and force and velocity performance across phases of the MC in resistance-trained, eumenorrheic women, though squat jump height performance was greater during the late follicular phase compared to the early follicular phase [16]. More recently, a meta-analysis looked at whether strength performance measures, including maximal voluntary contraction, isokinetic peak torque, and explosive strength differed across phases of the MC in eumenorrheic women [17]. From 21 studies, which included women that were classified as sedentary, recreationally active, moderately-to-well trained, and athletes, Blagrove et al. found no significant differences in strength measures between the early follicular, ovulatory, and mid luteal phases, making this an observation consistent with other meta-analyses [12,13,17]. Considering this body of evidence, performance variables encompassing a variety of muscular characteristics can vary throughout the MC, though this is an inconsistent finding and the magnitude of this variation appears to be small to trivial. Importantly, there is significant variability in the design of many of these studies, with inconsistencies in the methodology, as well as concerns with power and sample size, limiting the interpretation of how performance factors or variables could be impacted throughout the MC [12,13,17,18].

An important point worth noting is that two of these meta-analyses excluded studies in which women were using hormonal contraceptives, presumably, to narrow their investigations of how hormonal fluctuations might impact performance, which limits the generalization of their findings to women not using hormonal contraceptives [12,13]. With this, in a meta-analysis of 42 studies, Elliot-Sale et al. looked at the effects of oral contraceptive use on exercise performance (e.g., VO_2peak_, maximal and sub-maximal isometric force production, peak power and power decline during a Wingate test, etc.) compared to eumenorrheic women [9]. They found evidence that exercise performance could be negatively affected in women using oral contraceptives, though the effects were marginal. They also reported performance in these studies to be consistent throughout a hormonal contraceptive cycle in women using oral contraceptives, with no differences observed between days of using and not-using (placebo). Similar to the present study, Dasa et al. investigated whether strength and power performance metrics, including isometric grip strength, 20-m sprint time, jump height from a countermovement jump, and dynamic strength and power measures from leg press testing, differed between the follicular and luteal phases in competitive female athletes that were eumenorrheic or used hormonal contraceptives [19]. They found no differences in any of the metrics between the two phases, along with no differences between groups (eumenorrheic vs. hormonal contraceptives). In another study, Thompson et al. examined how muscular performance measures differed in women that were eumenorrheic or women using hormonal contraceptives that were categorized as either high- or low-androgenicity [20]. Participants were tested at three different times over one MC or hormonal contraceptive cycle. Consistent with most studies, they reported no significant differences in the performance measures across the three timepoints. They did note greater isokinetic knee flexion, bilateral hopping, and countermovement jump performance during the mid-luteal phase compared to the late follicular phase in the eumenorrheic group, as well as phasic differences in isokinetic knee flexion in the high-androgenicity group and countermovement jump performance in the low-androgenicity group.

Collectively, our findings are similar to those reported previously, indicating that metrics of muscular performance are largely unaffected by, or are similar, when tested at various points of the MC or a hormonal contraceptive cycle. Contrary to our original hypothesis, this suggests that the variability in estrogen and progesterone due to the MC or the use of hormonal contraceptives has a minimal impact on these performance measures. Despite not observing any significant variation between EF, OV, and ML, we found small to medium effect sizes for several variables when comparing performance between phases of the MC or HCC. For example, from the countermovement jump, we observed noteworthy effect sizes that favored EF in comparison to OV for several variables with clear performance-related implications, including relative max force (0.46), max velocity (0.7), and jump height (0.67). Similarly, we observed small to moderate effect sizes that favored ML in comparison to OV, including relative max force (0.34), max velocity (0.4), and braking rate of force development (0.77). Likewise, we observed small to moderate effect sizes for 15-yd sprint performance (EF vs. OV, ES = 0.35), peak velocity during the 70% back squat (OV vs. ML, ES = 0.27), bench press 1RM (EF vs. ML, ES = 0.33 and OV vs. ML, ES = 0.28), and 70% bench press RTF (EF vs. ML, ES = 0.39 and OV vs. ML, ES = 0.19). Our interpretation of these effect sizes is two-fold: (1) it provides greater context that performance did not favor OV in this study, which was our original hypothesis, and (2) it provides a degree of evidence that though performance did not significantly differ between phases, the magnitude of variability for some of the performance variables could be practically relevant. Within the context of sport and human performance, which can be concerned with variability in performance that is diminutive, these effect sizes could have greater implications, are likely relevant from a practical standpoint, and should be considered in greater detail in future studies. Further supporting potential variability, during each phasic visit to the lab (EF, OV, ML), we had participants complete a self-report assessment that addressed their overall readiness to perform testing each visit, including questions assessing their fatigue, motivation, energy levels, fatigue, and physiological symptoms (these were unpublished findings). From these assessments, these athletes reported greater readiness for performance testing (i.e., self-reported fatigue, energy levels, symptoms, and motivation scores that would favor being more ready to be tested) during OV and ML, counter to our findings for the performance measures. Despite the lack of significance, our observed effect sizes, when also considered with females often reporting perceived variability in performance throughout the MC or HCC, would seem to suggest there is a degree of difference in performance, which is relevant for females participating in high-caliber sport such as collegiate or professional athletes [21,22]. Against the context of previous literature on this topic, this disconnect between self-reported findings and performance, as well as the magnitude of variability of performance throughout the MC or HCC, requires further investigation. Additionally, future studies might consider determining smallest worthwhile change (SWC) thresholds which may provide more meaningful implications when working with competitive female athletes.

This study had several limitations as well as strengths. Recruitment of competitive female athletes to participate in research studies is challenging when factoring in their in-season schedules, training and practice schedules, along with their external commitments. With this, our study was limited to 11 female athletes, indicating this study was very likely underpowered. Given the effect sizes for some of our measures, with a larger sample, we may have been able to detect some significant differences, though this is speculatory. That this study was underpowered is relevant when considering our findings, particularly when considering some of the effect sizes we observed, as it poses a risk for Type II error. Our study relied on self-report, tracking, and day-counting methods to determine when participants were tested, which has been reported to be a sub-optimal methodological approach [18]. Recommendations for best practices when attempting to verify cycle phase include the combination of multiple methodologies, including calendar-based day counting, assessment of urinary surge in luteinizing hormone which typically occurs leading up to ovulation, and detection of estrogen and progesterone in serum using commercially available assays [18]. By using self-report, app tracking (FLO), and day-counting methods to establish phases of an MC or HCC, and not directly measuring hormone concentrations within the urine or blood, it limits our capacity to definitively compartmentalize phasic testing (EF, OV, ML) with these athletes in this study, and this is a frequently cited challenge within this field of research. This could also contribute to our lack of significant variability between the phases investigated in this study. Additionally, compared to other studies that analyzed differences between eumenorrheic women and women using hormonal contraceptives, we combined both users (Con) and non-users (Non) of hormonal contraceptives into a total pool to be analyzed. Con users were tested during the same cyclical timeframes as Non, which limits our ability to infer our findings for Con. However, as mentioned previously, performance measures appear to be consistent throughout a hormonal contraceptive cycle in these individuals, possibly making this a moot point [9,19,20]. Conversely, it is obvious that many women use (Con) and do not use (Non) hormonal contraceptives. Excluding one group limits the generalizability of findings. By including both groups, our study has good ecological validity. Worth noting, though it is not included in the formal analysis of this study, we performed exploratory, ex-post facto sub-group (Con vs. Non) effect size analysis of performance variables (15-yd sprint, back squat, 1RM bench press, and bench press RTF) throughout phases of the MC or HCC. We found (unpublished findings) a range of small to medium effect sizes (e.g., RTF for EF = 0.43, for OV = 0.66, for ML = 0.23), as well as large effect sizes (e.g., 1RM for EF = 1.38, for ML = 1.23), with performance favoring Con compared to Non for most variables assessed. Given our small sample size, it is difficult to extrapolate whether there is potential variability in performance between Con and Non throughout an HCC or MC. Nevertheless, this exploratory analysis is intriguing and creates a point of interest for future studies. Additionally, this study recruited collegiate female athletes—individuals with the most to gain from the applicable findings from this study. Women consistently report perceived differences in performance, readiness and/or preparedness for training or sport, symptoms, and fatigue throughout an MC or hormonal contraceptive cycle [1,21,22]. Thus, despite our findings and what the prevailing evidence says, women still perceive their performance differs. Our results have specific implications for competitive female softball players (collegiate, professional). Though not exclusively, many of the performance tests used in this study have practical relevance to the sport of softball, thus, this study is most generalizable to this population. Given this, our observations can help further facilitate the implementation of individualized training programming for softball players and similar athletes. It could be surmised that these athletes’ adaptation to and recovery from training is likely subject to similar degrees of variability, whether trivial or large in magnitude, which opens the discussion of extending the individualization of training programs to differences in positional demands and long-term programming needs (e.g., periodization). Finally, the practitioner should consider each athlete individually when factoring in the potential impact of cyclical variations in estrogen and progesterone on their physiology, training, recovery, and performance, while prioritizing their sleep, nutrition, psychological state, training load, and fatigue.

## 4. Materials and Methods

Collegiate softball players (n = 11; 19 ± 1 years) from Missouri State University were recruited for this study. Participants were excluded from this study if they showed sign/symptoms of cardiovascular, metabolic, or pulmonary disease, reported regularly using tobacco or related products, reported a history with amenorrhea or irregularities in their menstrual cycle, reproductive system disorders, or were pregnant or planned to become pregnant. Screening for exclusionary criteria was completed during a participant’s initial visit to the Exercise Physiology Lab. Following screening, 11 participants were eligible to participate in this study and completed the study’s requirements. All methodology and procedures used in this study were in accordance with the Declaration of Helsinki and were approved by Missouri State University’s Institutional Review Board (no. 113/10-24-2024).

### 4.1. Requirements, Protocols, and Timeline

Participants visited the lab four separate times across two consecutive MC or two consecutive HCC, with performance testing occurring during visits two, three, and four. During the initial visit, participants were familiarized with the study, completed an informed consent and a health history questionnaire that assessed the history, regularity, and length of their MC, along with their use of (Con) or non-use of (Non) hormonal contraceptives (oral, intrauterine devices). Participants then created an account to track and report their cycle history (FLO), which was used to determine the timing of participation for the remaining visits in the study. Participants’ body composition (InBody 770, InBody USA, Cerritos, CA, USA) and resting metabolic rate (TrueOne2400, Parvo Medics, Salt Lake City, UT, USA) were also measured during the initial visit. After completing the initial visit, participants (Non) were scheduled for the subsequent three visits corresponding to their respective EF, OV, and ML phases of the MC. Participant self-report and day-counting using the FLO app were used to determine the participation timeline for each participant. Given the effects of hormonal contraceptive use on phases and sub-phases of the menstrual cycle [5], participants that used hormonal contraceptives (Con) were scheduled for their three visits over a series of days of an HCC that was equivalent to participants not using hormonal contraceptives (Non) using the day-counting method.

Participants returned to the lab for performance testing over three separate visits that corresponded to the early follicular or early (EF), ovulatory or mid (OV), and mid luteal (ML) or late phases of their MC (Non) or HCC (Con). During each performance test during the visit, participants arrived in the morning in a rested and fasted state (0700-0900). Upon arrival, participants performed a countermovement jump on a force plate system (Sparta Science, Oura, Oulu, Finland). Participants then performed three, 15-yd sprints and sprint time was assessed using a timing gate system (Smart Speed, VALD, Newstead, Australia). Participants were given 90 s rest in between sprint attempts. Participants then performed three repetitions of the barbell back squat performed at 70% of their 1RM. Determination of velocity from back squat attempts was assessed using a linear position transducer (Vitruve Encoder, Madrid, Spain) and the attempt with the highest velocity was recorded. Participants then performed one-repetition maximum (1RM) testing for the bench press according to standard procedures for 1RM testing. Lastly, participants performed a repetitions to failure (RTF) test for the bench press using 70% of their established 1RM. Foall testing procedures, participants were given five minutes of rest between tests to minimize any residual fatigue from test-to-test. Participants repeated this sequence of testing for their other two visits. One athlete reported pre-existing injuries which limited their capacity to perform the 15-yd sprint and back squat. Thus, analysis of 15-yd sprint and back squat performance omitted this participant (n = 10), whereas the remaining variables included all participants (n = 11).

### 4.2. Statistical Analysis

Differences in data derived from the countermovement jump, 15-yd sprints, back squat, 1RM bench press, and bench press RTF between the EF, OV, and ML phases of the MC/HCC were analyzed using a repeated measures multiple analysis of variance (MANOVA). Bonferroni post hoc testing was used to make pairwise comparisons when applicable. Effect sizes (ES) were calculated and reported as the mean difference between phases divided by the pooled standard deviation for each variable. Data were reported as mean ± standard deviation and 95% confidence intervals (CI), with alpha (α) set at *p* ≤ 0.05. An a priori power analysis was conducted using G*Power version 3.1.9.7 to determine the appropriate sample size for detecting significant differences in the dependent variables in this study. The power analysis was subject to the null and alternative hypotheses in this study. The null hypothesis (H_0_) was as follows: There is no significant difference in performance measures between EF, OV, and ML, whereas the alternative hypothesis (H_1_) was as follows: There is a significant difference in performance measures between EF, OV, and ML. To achieve 80% power (1 − β, 0.8, β = 0.2) for detecting a moderate effect size (0.5), at a significance level of α = 0.05, the recommended sample size was n = 24. All statistical analyses were performed using SPSS statistical software (IBM SPSS v.28.0, Armonk, NY, USA).

## 5. Conclusions

There were no significant differences following a series of tests that assessed several muscular performance characteristics between three phases or timepoints of the MC or HCC in collegiate female athletes that use (Con) or do not use (Non) hormonal contraceptives. Though not significant, most variables derived from the countermovement jump, back squat velocity, along with upper extremity maximal strength and endurance tended to be highest during the ML phase or later phase of the hormonal contraceptive cycle, with additional variables derived from the countermovement jump being highest during the EF phase or early phase of the hormonal contraceptive cycle. This study supports existing literature that shows muscular performance characteristics are largely unaffected by cyclical variations in estrogen and progesterone throughout the course of an MC or HCC—though some variation does occur. However, we did observe small to moderate effect sizes for performance variables between phases of the MC/HCC, providing evidence that though there was no significant variability, the magnitude of variability is noteworthy. Additionally, it is important to reiterate that this study was underpowered given its sample size, which increases the risk for Type II error and potentially masks any observed differences. Many athletes report readiness, preparedness, and performance differences throughout an MC or HCC which could affect performance during exercise, training, or sport. The practitioner must take this into consideration when designing and implementing training, programming, and recovery strategies and tailor these to each athlete’s needs.

## Figures and Tables

**Figure 1 muscles-04-00037-f001:**
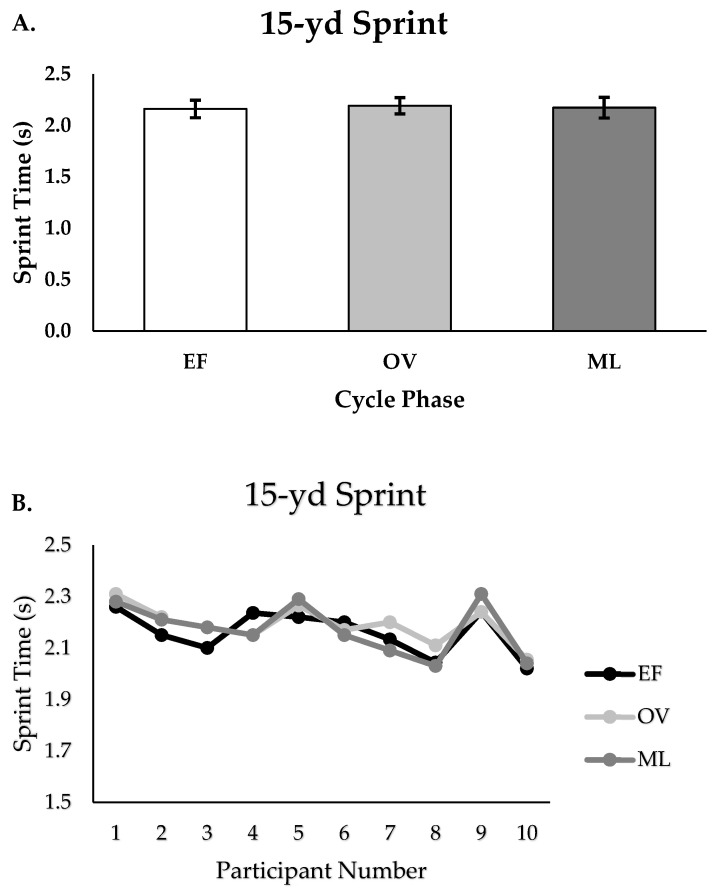
(**A**,**B**). Fifteen-yard sprint performance for group (n = 10) and individualized performance for EF, OV, and ML phases (*p* = 0.839); Group data are reported as mean ± SD; EF = early follicular or early, OV = ovulatory or mid, ML = mid luteal or late.

**Figure 2 muscles-04-00037-f002:**
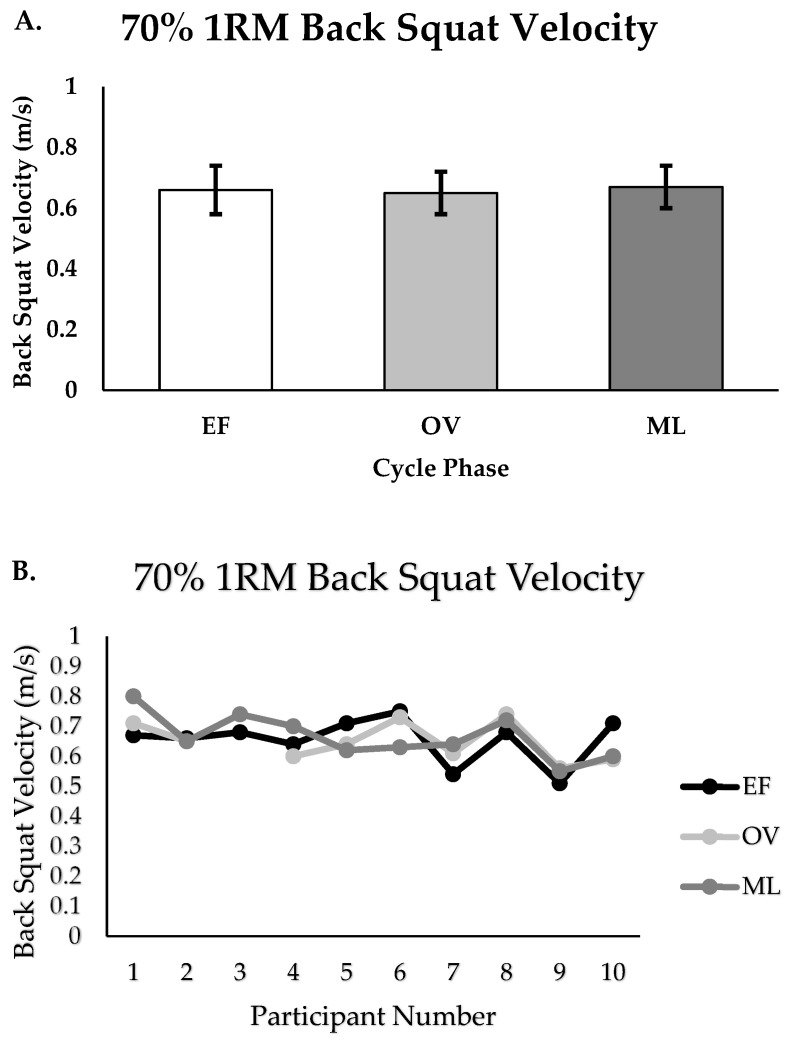
(**A**,**B**). Back squat velocity for group (n = 10) and individualized performance for EF, OV, and ML phases (*p* = 0.871); Group data are reported as mean ± SD; EF = early follicular or early, OV = ovulatory or mid, ML = mid luteal or late.

**Figure 3 muscles-04-00037-f003:**
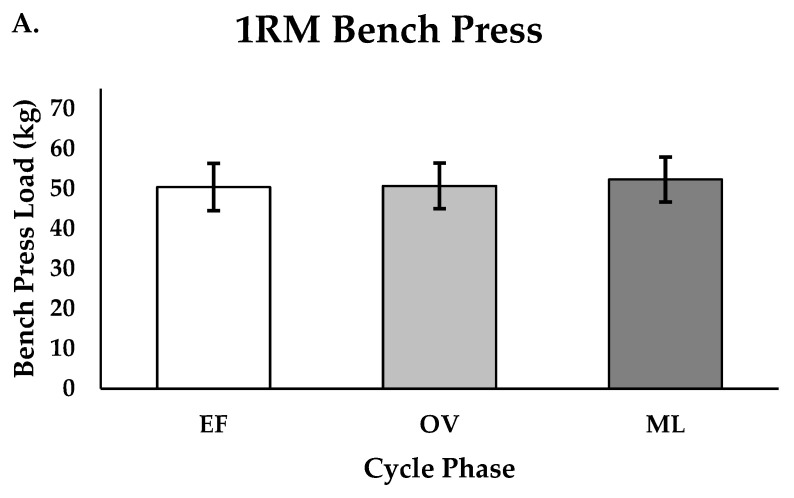
(**A**,**B**). Bench press 1RM for group (n = 11) and individualized performance for EF, OV, and ML phases (*p* = 0.717); Group data are reported as mean ± SD; EF = early follicular or early, OV = ovulatory or mid, ML = mid luteal or late.

**Figure 4 muscles-04-00037-f004:**
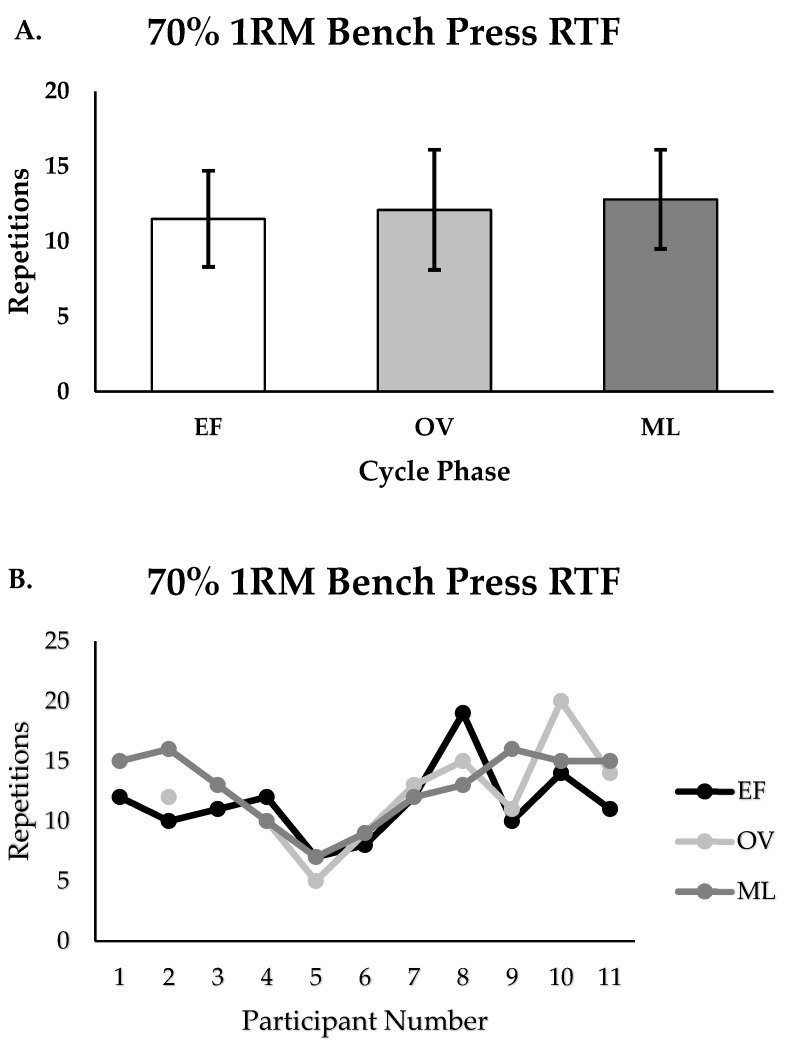
(**A**,**B**). Bench press repetitions to failure (RTF) for group (n = 11) and individualized performance for EF, OV, and ML phases (*p* = 0.645); Group data are reported as mean ± SD; EF = early follicular or early, OV = ovulatory or mid, ML = mid luteal or late.

**Table 1 muscles-04-00037-t001:** Participant descriptive characteristics at initial visit.

	Age (Years)	Height (cm)	Mass (kg)	BMI (kg/m^2^)	Body Fat (%)	Muscle Mass (kg)	Resting Metabolic Rate (kcal)
Non (n = 7)	19 ± 1	169 ± 4.4	68 ± 5	24.6 ± 2.6	25.1 ± 3.7	28.4 ± 2.0	1583 ± 106
Con (n = 4)	19 ± 1	167.1 ± 8.6	73.4 ± 13.2	26.1 ± 2.0	23.8 ± 1.0	31.6 ± 6.0	1679 ± 225
Total (n = 11)	19 ± 1	168.4 ± 5.9	69.9 ± 8.7	25.1 ± 2.4	24.6 ± 3.0	29.6 ± 4.0	1618 ± 156
					(22.8–26.4)	(27.2–31.9)	(1525–1710)

Data presented as mean ± SD; n = 11; and 95% CI values for body fat, muscle mass, and resting metabolic rate are listed below the mean and SD for each variable in parentheses; Con = hormonal contraceptive-users, Non = non-contraceptive-users.

**Table 2 muscles-04-00037-t002:** Countermovement jump variable performance and effect sizes between EF, OV, and ML.

	EF	OV	ML	Effect Size: EF vs. OV/EF vs. ML/OV vs. ML
Relative Max Force (N/kg)	17.1 ± 1.0	16.7 ± 0.7	17.1 ± 1.5	EF vs. OV = 0.46; EF vs. ML = 0; OV vs. ML = −0.34
Relative Max Power (W/kg)	28.9 ± 4.2	28.0 ± 4.1	27.3 ± 4.5	EF vs. OV = 0.22; EF vs. ML = 0.37; OV vs. ML = 0.16
Max Velocity (m/s)	2.64 ± 0.1	2.57 ± 0.1	2.61 ± 0.1	EF vs. OV = 0.7; EF vs. ML = 0.3; OV vs. ML = −0.4
Concentric time (s)	0.34 ± 0.04	0.35 ± 0.03	0.34 ± 0.04	EF vs. OV = −0.28; EF vs. ML = 0; OV vs. ML = 0.28
Eccentric time (s)	0.41 ± 0.1	0.38 ± 0.1	0.4 ± 0.1	EF vs. OV = 0.3; EF vs. ML = 0.1; OV vs. ML = −0.2
Jump height (m)	0.36 ± 0.03	0.34 ± 0.03	0.35 ± 0.03	EF vs. OV = 0.67; EF vs. ML = 0.33; OV vs. ML = −0.33
Time to Max Acceleration (s)	0.99 ± 0.2	0.91 ± 0.2	0.91 ± 0.1	EF vs. OV = 0.4; EF vs. ML = 0.51; OV vs. ML = 0
Time to Takeoff (s)	1.10 ± 0.2	1.02 ± 0.2	1.03 ± 0.1	EF vs. OV = 0.4; EF vs. ML = 0.44; OV vs. ML = −0.06
Braking RFD (N/s)	2294.8 ± 1361.8	2259.1 ± 1286.0	2478.2 ± 1156.2	EF vs. OV = 0.03; EF vs. ML = −0.15; OV vs. ML = 0.77

Data presented as mean ± SD; n = 11; EF = early follicular, OV = ovulatory, ML = mid-luteal; N/kg = newtons per kilogram, W/kg = watts per kilogram, m = meters, s = seconds, m/s = meters per second, N/s = newtons per second.

## Data Availability

The data presented in this study are available upon request from the corresponding author.

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
