# Peer review of "Muscular Performance Is Not Significantly Altered Throughout Phases of the Menstrual Cycle or a Hormonal Contraceptive Cycle in Collegiate Softball Players"

_muscles, 2025, doi:10.3390/muscles4030037_

Round 1

Reviewer 1 Report

Comments and Suggestions for Authors

Below are the reviewer’s comments intended to improve the overall quality of the manuscript.

The title is clear and informative, but the phrase "remains constant" may be too strong considering the small sample size and methodological limitations. It would be more appropriate to use a more cautious expression such as "shows no significant variation" to better reflect the study’s findings.

Abstract

Although the results were not statistically significant, the abstract should still report the relevant statistical test values (e.g., p-values, effect sizes). Additionally, the concluding statement should be rephrased, as it currently appears too definitive given the small sample size of only 11 participants. A more cautious interpretation is recommended to accurately reflect the study’s limitations.

I have several major remarks regarding the manuscript. The topic is of interest and the overall structure is well organized; however, the most prominent limitation is the small sample size.

  1. This substantially reduces statistical power. Although the authors acknowledge this, it should be emphasized more explicitly in both the abstract and the conclusions as a key limitation.
  2. While a priori power analysis is mentioned, it is not presented clearly for the specific tests conducted. Providing these details would improve transparency.
  3. The method for determining menstrual cycle phases relied solely on the FLO application and self-reporting, which carries a risk of phase misclassification. I recommend briefly discussing in the manuscript how such potential misclassifications may have influenced the results. The absence of hormonal verification (e.g., blood sampling) further limits precision.
  4. The statistical interpretation is primarily based on the absence of significant differences. It would be beneficial to incorporate a more thorough discussion of the observed effect sizes and their potential practical relevance.

Combining data from women using and not using hormonal contraceptives limits the ability to draw definitive

Author Response

Comment 1: The title is clear and informative, but the phrase "remains constant" may be too strong considering the small sample size and methodological limitations. It would be more appropriate to use a more cautious expression such as "shows no significant variation" to better reflect the study’s findings.

Response 1: This is a good point and we changed the title to reflect this. Thanks for the suggestion. 

Comment 2: Although the results were not statistically significant, the abstract should still report the relevant statistical test values (e.g., p-values, effect sizes). Additionally, the concluding statement should be rephrased, as it currently appears too definitive given the small sample size of only 11 participants. A more cautious interpretation is recommended to accurately reflect the study’s limitations.

Response 2: This is a fair point. Muscles has an abstract character/word count limitation on the abstract, so we opted to not include the p values or effect sizes in the abstract, while also allowing the abstract to describe the study design in detail. There was a large amount of variables included in the dataset and to include them would put us far past the acceptable character count. We believe how it is written, though brief and broad, summarizes our findings accurately and succinctly. To the second part, we agree and this is a good point. We have made modifications to the end of the conclusion to make it less definitive given the small sample size. 

Comment 3: I have several major remarks regarding the manuscript. The topic is of interest and the overall structure is well organized; however, the most prominent limitation is the small sample size... This substantially reduces statistical power. Although the authors acknowledge this, it should be emphasized more explicitly in both the abstract and the conclusions as a key limitation.

Response 3: Thank you for this comment. We agree. We have added additional text to the abstract and conclusion sections that reiterates that this study was underpowered and that this should be taken into consideration when examining this paper's findings. 

Comment 4: I have several major remarks regarding the manuscript. The topic is of interest and the overall structure is well organized; however, the most prominent limitation is the small sample size... While a priori power analysis is mentioned, it is not presented clearly for the specific tests conducted. Providing these details would improve transparency.

Response 4: We have made modifications to the Statistical Analysis sub-section that expands on the power analysis as part of this study. 

Comment 5: I have several major remarks regarding the manuscript. The topic is of interest and the overall structure is well organized; however, the most prominent limitation is the small sample size... The method for determining menstrual cycle phases relied solely on the FLO application and self-reporting, which carries a risk of phase misclassification. I recommend briefly discussing in the manuscript how such potential misclassifications may have influenced the results. The absence of hormonal verification (e.g., blood sampling) further limits precision.

Response 5: Thank you for this comment. We agree. We have added additional text to the discussion section that addresses this as a limitation of our study. 

Comment 6: I have several major remarks regarding the manuscript. The topic is of interest and the overall structure is well organized; however, the most prominent limitation is the small sample size... The statistical interpretation is primarily based on the absence of significant differences. It would be beneficial to incorporate a more thorough discussion of the observed effect sizes and their potential practical relevance.

Response 6: Excellent point. We have discussed some of our effect sizes we observed to a greater detail in the discussion and conclusion sections. We also addressed the potential implications of these effect sizes in the discussion section. 

Reviewer 2 Report

Comments and Suggestions for Authors

The manuscript explores the influence of menstrual cycle (MC) and hormonal contraceptive cycle (HCC) phases on muscular performance in collegiate softball players. The topic is relevant and aligns with the journal’s scope, offering applied insights for sports practitioners and coaches working with female athletes. The integration of both contraceptive users and non-users enhances ecological validity. The presentation is clear, and the literature review is well-structured.

However, several key areas require improvement to strengthen the study’s scientific contribution:

  1. Sample Size and Statistical Power – The study is underpowered (n = 11 vs. n = 24 estimated in the a priori calculation). This limitation should be emphasized more strongly in the discussion and conclusions, with a cautionary note about the risk of Type II errors. Given the effect size patterns, the discussion should focus not only on null results but also on practical significance.

  2. Phase Identification Methodology – Reliance on self-report, app tracking, and day-counting introduces classification uncertainty. The limitations section should explicitly address how this could attenuate differences, and reference gold-standard methods (hormonal assays, ovulation testing).

  3. Pooled Analysis of Contraceptive and Non-Contraceptive Users – Combining these groups may mask physiological differences. Even if underpowered, an exploratory subgroup analysis with effect sizes (not only p-values) could offer valuable insight.

  4. Data Presentation – Consider adding individual data plots (e.g., spaghetti plots) to better illustrate within-subject variability, which is particularly relevant for MC research. Clarify effect size interpretation and, if possible, add smallest worthwhile change (SWC) thresholds for practical context.

  5. Interpretation of Findings – Reframe “no significant differences” to acknowledge possible small, practically relevant effects. Link this to athlete perceptions and readiness, potentially adding a short summary of subjective reports from participants.

  6. Practical Implications – The applied recommendations would benefit from being more sport-specific (softball context) and linking directly to periodization, positional demands, and conditioning strategies.

Overall, the manuscript is well-presented and tackles an important area in female athlete physiology. With the suggested revisions, it would make a stronger and more nuanced contribution to the literature.

Author Response

Comment 1: Sample Size and Statistical Power – The study is underpowered (n = 11 vs. n = 24 estimated in the a priori calculation). This limitation should be emphasized more strongly in the discussion and conclusions, with a cautionary note about the risk of Type II errors. Given the effect size patterns, the discussion should focus not only on null results but also on practical significance.

Response 1: Excellent point that was also mentioned by the other reviewer. We have addressed in this in some detail in the discussion and conclusion sections. We have also expanded on our interpretation of the observed effect sizes in the discussion section, while highlighting this again in the practical application portion towards the end of the discussion. 

Comment 2: Phase Identification Methodology – Reliance on self-report, app tracking, and day-counting introduces classification uncertainty. The limitations section should explicitly address how this could attenuate differences, and reference gold-standard methods (hormonal assays, ovulation testing).

Response 2: Excellent point and is one that was also mentioned by the other reviewer. We have discussed this to a greater extent in the discussion, with particularly emphasis on how our methodology compares to current best practices, as well as the implications this could have on our findings. 

Comment 3: Pooled Analysis of Contraceptive and Non-Contraceptive Users – Combining these groups may mask physiological differences. Even if underpowered, an exploratory subgroup analysis with effect sizes (not only p-values) could offer valuable insight.

Response 3: This is a valuable point. This was not an aim of this study; however, it raises an interesting point worth considering. We analyzed differences in effect size at the sub-group level (Non vs Con) for several of our performance variables and included a "snippet" of our observations in the discussion section. We made it clear these were unpublished findings. We did not include this in the formal analysis due to the small sample size but addressed it in the discussion. 

Comment 4: Data Presentation – Consider adding individual data plots (e.g., spaghetti plots) to better illustrate within-subject variability, which is particularly relevant for MC research. Clarify effect size interpretation and, if possible, add smallest worthwhile change (SWC) thresholds for practical context.

Response 4: This is a valuable point. We created and input figures for 4 of the performance tests (sprint, back squat, 1RM, 70% RTF) that shows individualized data points in addition to the group mean/sd. Given our data set, effect sizes, sample size, and scope of the paper, we did not include SWC thresholds. However, we included this as a future perspective worth considering in the discussion section. 

Comment 5: Interpretation of Findings – Reframe “no significant differences” to acknowledge possible small, practically relevant effects. Link this to athlete perceptions and readiness, potentially adding a short summary of subjective reports from participants.

Response 5: Excellent point. We addressed this in the discussion section by expanding on our effect sizes and discussing their implications. 

Comment 6: Practical Implications – The applied recommendations would benefit from being more sport-specific (softball context) and linking directly to periodization, positional demands, and conditioning strategies.

Response 6: Fair point. We added a short section to the discussion section that addresses this. 

Round 2

Reviewer 1 Report

Comments and Suggestions for Authors

 thanks for addressing the comments

Author Response

Thank you!

Reviewer 2 Report

Comments and Suggestions for Authors

The revised version of your manuscript shows clear improvements compared to the initial submission. You have adequately addressed the main reviewer concerns, including: (1) explicit discussion of statistical power and Type II error risk in both the Discussion and Conclusions, (2) clarification of phase identification methodology and reference to gold-standard approaches, (3) inclusion of exploratory subgroup analyses (contraceptive vs. non-users), (4) a more cautious interpretation of non-significant findings with emphasis on effect sizes and smallest worthwhile change thresholds, and (5) integration of athlete self-reported readiness data.

The presentation is clear and well-organized, with updated references and stronger applied recommendations for softball athletes. Only minor issues remain: figures could be further improved by adding individual traces (spaghetti plots), some discussion sections could be streamlined to avoid redundancy, and the reference list should be checked for potential duplication (Elliott-Sale et al. 2020).

Overall, the manuscript is now substantially strengthened and, after minor revisions, will provide a valuable contribution to the literature on female athlete performance across menstrual and contraceptive cycles.

Author Response

Comments 1: The presentation is clear and well-organized, with updated references and stronger applied recommendations for softball athletes. Only minor issues remain: figures could be further improved by adding individual traces (spaghetti plots), some discussion sections could be streamlined to avoid redundancy, and the reference list should be checked for potential duplication (Elliott-Sale et al. 2020).

Response 1: Thank you for your comments and suggestions, they do increase the quality of the manuscript. We have modified the individual data figures in a way that we believe more effectively demonstrates individual variability between EF, OV, and ML. This is shown in Fig 1B, 2B, 3B, and 4B. We removed the duplication in the references - excellent find! - and we modified the in-text citation list throughout where it was applicable. Finally, we read through the discussion and removed redundancies where we found them.